# Traffic trajectory data analysis technology based on HMM model map matching algorithm

Mingkang Sun[1]*, Xiang Li[2]

1 Glasgow College, University of Electronic Science and Technology of China, Chengdu, 610000, China,
2 Siemens Industrial Automation Products Ltd. Chengdu, Chengdu, 611730, China

* ericasunmk@163.com

## Abstract

The rapid growth of traffic trajectory data and the development of positioning technology have driven the demand for its analysis. However, in the current application scenarios, there are some problems such as the deviation between positioning data and real roads and low accuracy of existing trajectory data traffic prediction models. Therefore, a map matching algorithm based on hidden Markov models is proposed in this study. The algorithm starts from the global route, selects K nearest candidate paths, and identifies candidate points through the candidate paths. It uses changes in speed, angle, and other information to generate a state transition matrix that match trajectory points to the actual route. When processing trajectory data in the experiment, K = 5 is selected as the optimal value, the algorithm takes 51 ms and the accuracy is 95.3%. The algorithm performed well in a variety of road conditions, especially in parallel and mixed road sections, with an accuracy rate of more than 96%. Although the time loss of this algorithm is slightly increased compared with the traditional algorithm, its accuracy is stable. Under different road conditions, the accuracy of the algorithm is 98.3%, 97.5%, 94.8% and 96%, respectively. The accuracy of the algorithm based on traditional hidden Markov models is 95.9%, 95.7%, 95.4% and 94.6%, respectively. It can be seen that the accuracy of the algorithm designed has higher precision. The experiment proves that the map matching algorithms based on hidden Markov models is superior to other algorithms in terms of matching accuracy. This study makes the processing of traffic trajectory data more accurate.

## 1. Introduction

Traditional map matching (TMM) algorithms, primarily relying on geometric proximity, excel in simplicity and computational efficiency [1]. They perform well in scenarios with high-quality data and simple road network structures, where geometric factors are sufficient to determine the correct path [2,3]. However, these methods fall short in complex urban environments where dense road networks and frequent intersections introduce ambiguity [4,5]. The conventional approach also fails to consider the temporal sequence of the trajectory data. In

**Data Availability Statement:** All relevant data are within the manuscript and its Supporting Information files.

**Funding:** The author(s) received no specific funding for this work.

**Competing interests:** The authors have declared that no competing interests exist.

contrast, the Hidden Markov model (HMM)-based map matching algorithm offers a robust alternative by incorporating not only the geometric data but also the temporal sequence and spatial topology of the road network. This probabilistic model is particularly adept at handling the noise and imprecision inherent in GPS data, providing a more accurate reflection of the actual vehicle path [6,7]. To address the deviation between positioning data and real roads, a map matching algorithm based on HMM has been proposed. This algorithm takes a global perspective and utilizes changes in speed, angle, and other information to achieve matching between location data and real roads. The research structure mainly includes four parts. Firstly, it summarizes the research achievements and shortcomings of trajectory data and HMM algorithm at home and abroad. Secondly, it researches and designs HMM models and implements map matching algorithms. Then, trajectory data are compared and analyzed through experiments. Finally, the experimental results are summarized, the shortcomings of the research are pointed out, and future research directions are proposed.

## 2. Related works

With the development of urbanization in China, processing traffic trajectory data is crucial. Map matching is one of the important links in traffic trajectory data analysis, which maps trajectory data to corresponding locations on the map. This field has attracted the attention of massive scholars. To estimate travel time from sparse or low resolution trajectory data, K. Zhang et al. designed a generative adversarial network algorithm which considered spatio-temporal correlation and generated travel time for road segments without sufficient observation data. The experimental results showed that this algorithm could effectively estimate the average travel time of the link under various data loss rates, and was superior to other algorithms. In addition, the accuracy of the algorithm reached more than 98%, and the average time was 120ms. However, the weakness of this study is that the algorithm is often not stable enough, and is prone to pattern oscillation and gradient disappearance [8]. To construct the spatial semantic road map, J. Huang et al. proposed a divide and conquer approach, which integrates multiple data sources. This method includes two tasks: road structure reconstruction and road attribute inference. Example analysis showed that in the application of Wuhan City, this method could construct a routable road map with enhanced geometric structure and rich semantic information. In addition, the spatial semantic road map constructed could provide data support for vehicle navigation and spatial data infrastructure updates, and had good quality. However, this study also has certain shortcomings, such as difficulty in data integration, data security issues, and management costs [9]. Regarding the calibration issue of the car following model, R. Keane and other researchers compared different solving algorithms and finally determined an effective optimization method. This method could calibrate any car following model to adapt to various trajectory data, including lane changes, etc. The results indicated that the quasi Newtonian algorithm using the adjoint method was faster than the genetic algorithm and achieved higher calibration model accuracy. Besides, the accuracy of the quasi Newton algorithm using adjacency method was as high as 97.52%, and the computational cost was relatively low. However, this study also has shortcomings, such as increasing the complexity of the problem, requiring a large amount of computation, and not being suitable for solving non smooth problems [10]. To generate a qualified trajectory dataset that is difficult to distinguish from real trajectories, X. Chen et al. proposed two advanced solutions, TrajGAN and TrajVAE. These methods used long short-term memory models to model trajectory features and generated trajectory data using generative adversarial networks and variational autoencoder frameworks. The experimental results showed that the average accuracy values of the TrajGAN and TrajVAE models were 97.82% and 98.87%, respectively, and the generated

trajectories were very close to the existing trajectories. These methods are more accurate and stable than baseline models. However, there are also shortcomings in this study, such as the tendency for gradient vanishing, difficulty in controlling the quality of generated samples, sensitivity to parameter settings, and high demand for computing resources [11].

To solve the efficiency degradation of TMM methods based on HMM when facing dense road networks, G. Cui and other experts designed a line segment-based HMM map matching model. This model requires dividing the GPS trajectory into multiple sub-trajectories. The results indicated that the proposed method improved the effectiveness and efficiency of TMM methods. The accuracy of this method was over 92%, with an average time of 213ms and an F1 value of 0.973. However, this study also needs to be improved, such as a high demand for computing resources [12]. M. Dogramadzi and other scholars proposed a map matching algorithm based on HMM to reduce measurement errors in travel trajectory data. This algorithm could accelerate the calculation of transition probability through high data availability instances. The results showed that this model was able to reduce the running time by about 0.90 compared to the traditional HMM. The accuracy of this model could reach up to 99.21%, which can reduce the demand for processor and memory usage, and accelerate the calculation of transition probabilities. However, this study can further improve the simulation of GPS errors [13]. To solve the positioning errors in map matching, P. Alrassy and other experts proposed a map matching algorithm with traffic data and trajectory data and combined a map matching framework based on probability and weight. In addition, the ground reality data collected by the vehicle sensor device was analyzed and compared with the commonly used off-the-shelf map matching platform. The results showed that the algorithm had strong robustness, with an accuracy of 97.45%, especially when map data is dense and GPS noise is high. In addition, this algorithm can perform map matching quickly and accurately. However, this study also has certain shortcomings, such as a lack of in-depth exploration in data security and privacy, and insufficient consideration in the practical implementation [14]. To reduce the impact of positioning errors and accelerate the matching process of high sampling rate trajectories, L. Huang et al. proposed a batch matching strategy model for track data sub-sequences. To ensure that the sub-sequences to be matched are on the same road section, the speed estimation was used as a constraint to determine the sub-sequence size and the local features of the road network. The results indicated that this algorithm was superior to algorithms designed for low sampling rate trajectories. This algorithm has good efficiency and accuracy, and can effectively reduce the impact of positioning errors. However, there are still significant shortcomings in the incremental map matching algorithm, and further research is needed [15].

In summary, scholars and experts have designed some trajectory data and Markov models for various fields. However, there is rare research on traffic trajectory data and applying it to map matching algorithms. Therefore, this study proposes an HMM-based map matching algorithm, which introduces HMM model and time series information, comprehensively considers the semantic information of adjacent time intervals of trajectory points to infer the true location. The goal is to solve problems in traffic trajectory data analysis and improve the accuracy and practicality of the map matching algorithm.

## 3. Methodology

Traditional technical methods usually start from a geometric perspective and use methods such as projection and point-to-point distance for matching [16]. However, as the complexity of roads increases, the matching accuracy of traditional technical means gradually decreases. Therefore, a map matching algorithm based on HMM is proposed.

## 3.1 Design of HMM and trajectory matching

Trajectory matching matches the obtained trajectory data with the actual road using a specific algorithm. Due to the limited accuracy of existing positioning instruments, there may be some deviation between the data collected by the instruments and the actual road conditions. Therefore, it is necessary to match the data with the actual road conditions [17,18]. Track data $T$ is a series of data generated in chronological order, including longitude, dimensionality, and time information, as shown in Eq (1) [19].

$$L = \{p_1, p_2, \cdots, p_n\} \tag{1}$$

In Eq (1), $p_i \in L$ contains three elements, as shown in Fig 1. The time difference between $p_i$ and $p_{i+1}$ is only $t$.

From Fig 1, a series of data between P1 and Pn have different trajectories and information. The longitude, dimension, and time information of a series of data between P1 and Pn are shown in Table 1.

From Table 1, the longitude of the data from P1 to P8 is around 30.70, the latitude is around 104.0 and 104.1, and the time is around 9:30 am. The determination of trajectory data $T$ is mainly obtained through sampling instruments. Road segment $e$ can be cut from a real section of the road according to certain rules, where $e_{id}$ means the $id$ of that section of the road. $e_{id,start}$ represents the beginning of the section of road. $e_{id,end}$ denotes the end of this section of road, as shown in Fig 2.

Network route map $G(V, E)$ represents the network route map to be processed, where $V$ contains the start and end points of all road segments. $E$ represents all line segments. Line segments are elements in a network circuit diagram that represent paths or connections in a network [20,21]. The specific form is shown in Fig 3.

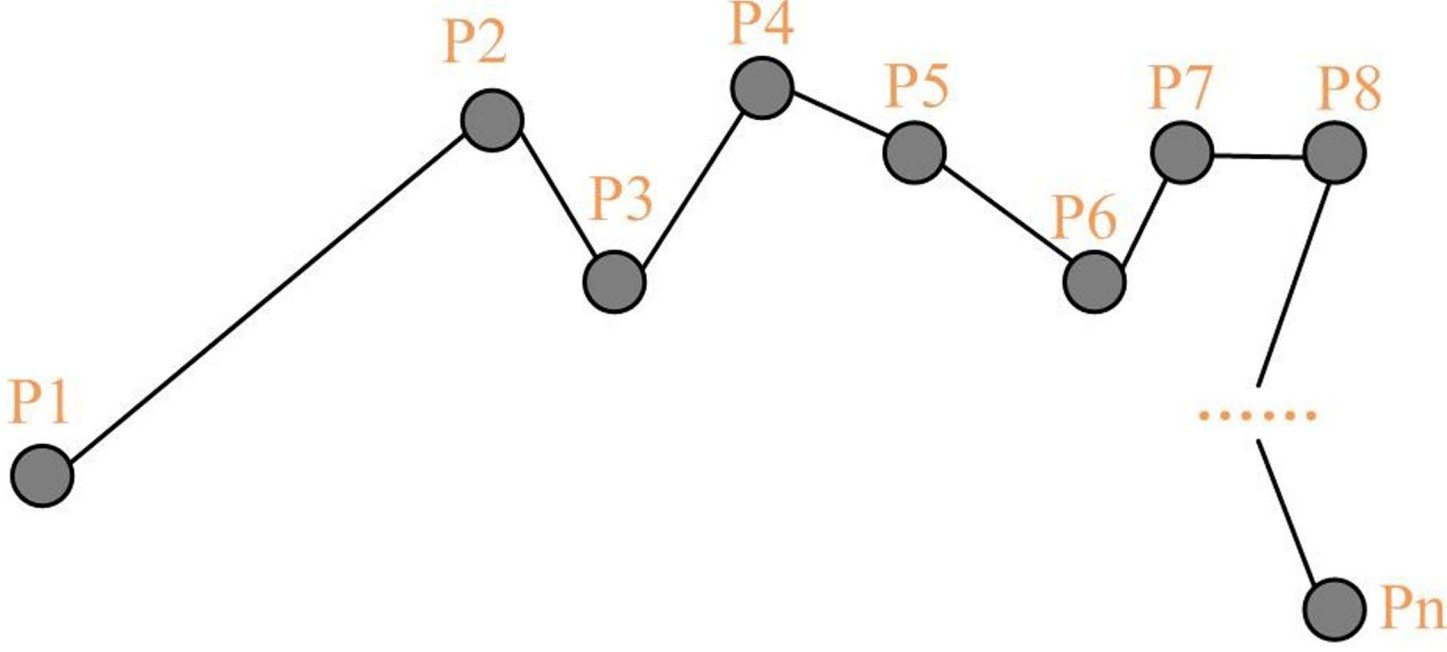

**Fig 1. Trajectory data diagram.**

**Table 1. The longitude, dimension, and time information of a series of data between P1 and Pn.**

| Data | Latitude | Longitude | Time |
|------|----------|-----------|------|
| P1 | 104.10041 | 30.70883 | 9:24 |
| P2 | 104.10018 | 30.70847 | 9:27 |
| P3 | 104.10002 | 30.70823 | 9:30 |
| P4 | 104.09996 | 30.70811 | 9:33 |
| P5 | 104.09988 | 30.70794 | 9:36 |
| P6 | 104.09983 | 30.70788 | 9:39 |
| P7 | 104.09979 | 30.70782 | 9:42 |
| P8 | 104.09975 | 30.70766 | 9:45 |
| . . . . . . | . . . . . . | . . . . . . | . . . . . . |
| Pn | . . . . . . | . . . . . . | . . . . . . |

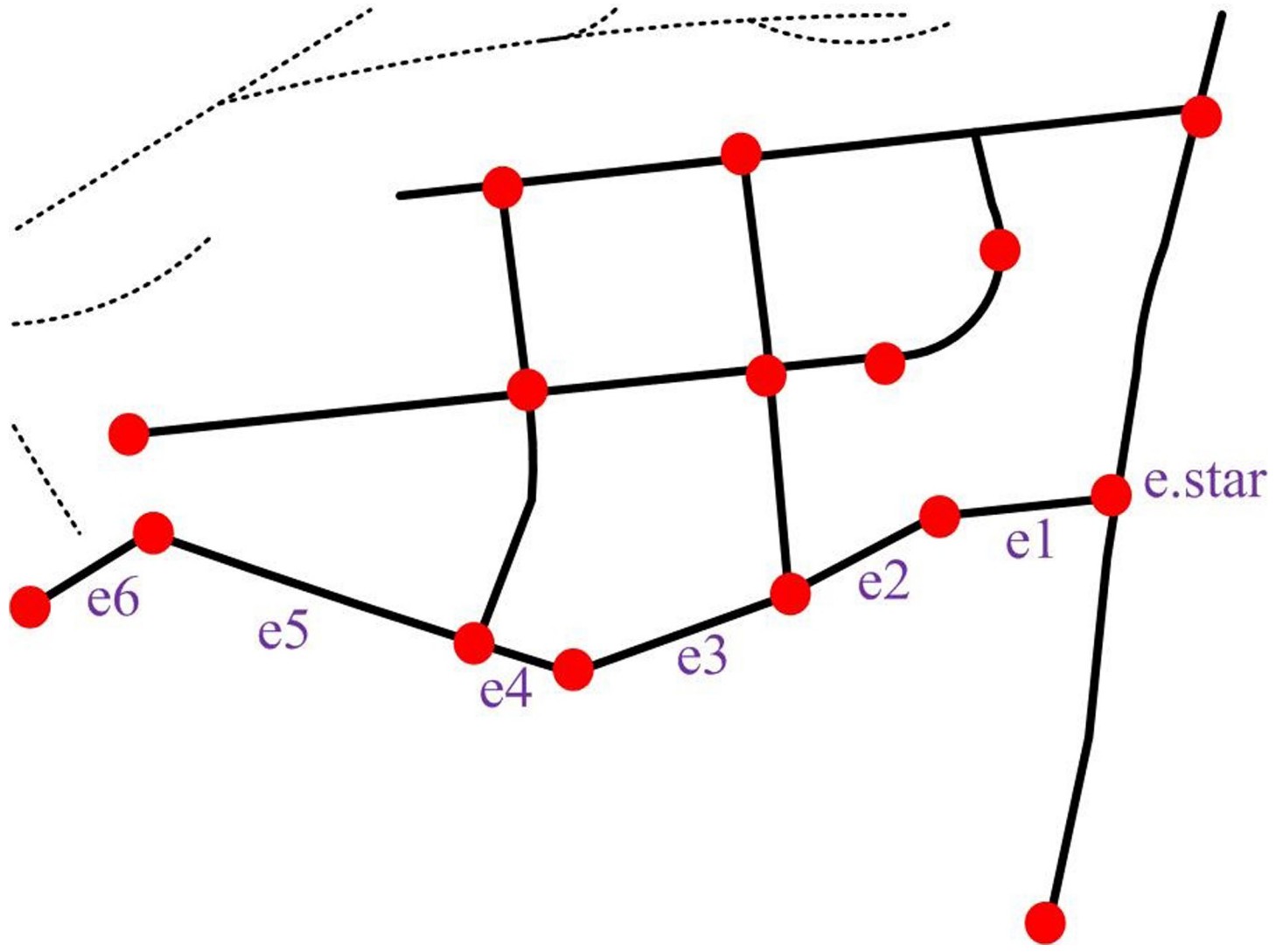

**Fig 2. Road segment group map.**

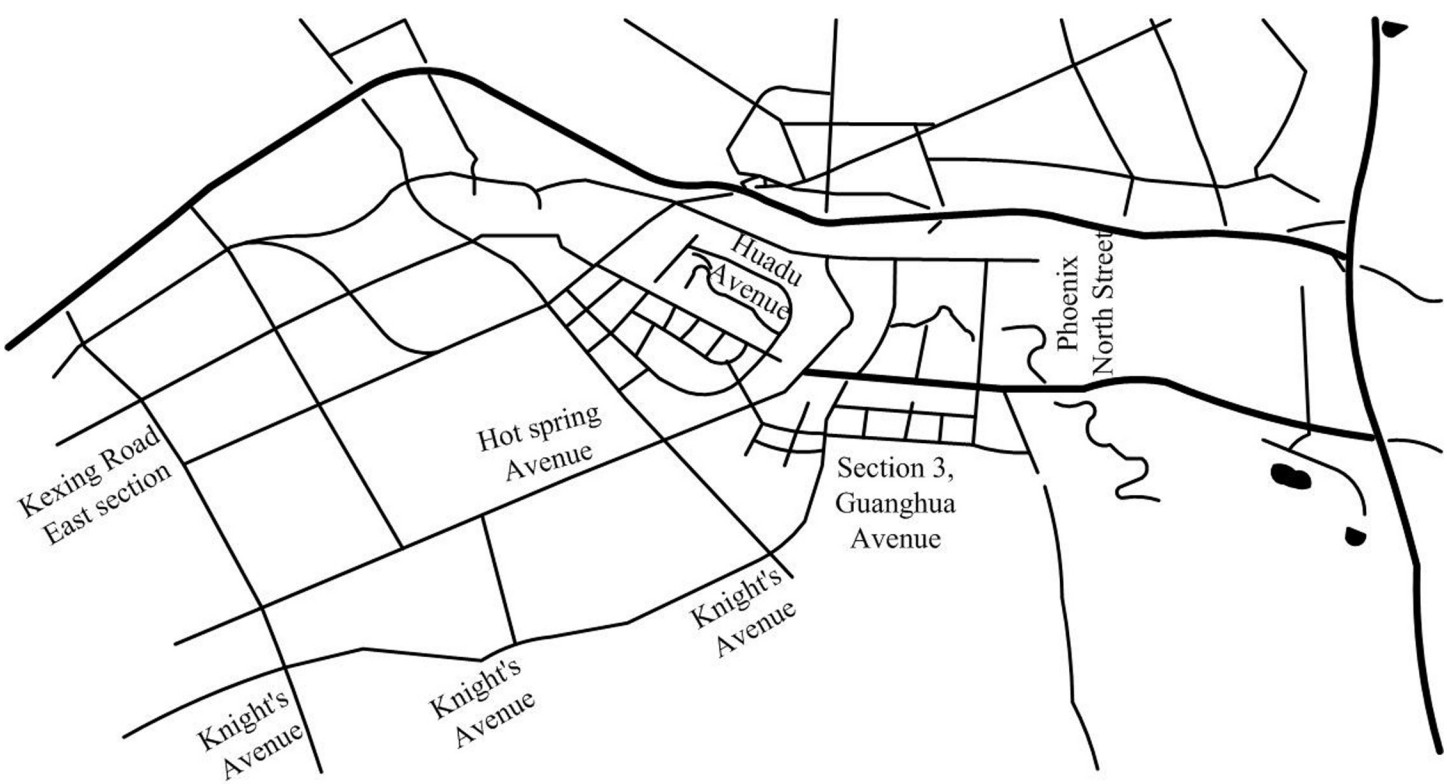

**Fig 3. Road network map.**

By providing a trajectory data *T*, it searches for a set of real *e* in network line G to represent the trajectory data. This algorithm for finding real data is called map matching. However, geometric map matching algorithms often face problems such as noise and uncertainty, deviation from paths, data sparsity, dynamic changes, and multiplicity [22,23], as shown in Fig 4.

Fig 4 shows a schematic diagram of map matching using track data. Simply considering the geometric factors without considering the logic in the time series of the trajectory point, the point will be located on line B, but its true matching point is on line A. This shows that if matching is based only on geometric proximity, the algorithm may ignore the logic in the time series of the trajectory points. The complexity of urban road networks may also lead to the inaccuracy of pure geometric matching methods. Intersections, parallel roads, overpasses and other factors can make the closest distance not always represent the correct match. Therefore, a map matching algorithm considering the logic of time series was adopted.The algorithm can match the map more accurately by introducing HMM, considering the geometry, and combining the time series information of track points. The HMM model can process data with temporal characteristics, that is, there is a chronological order between the data. The key concepts include observed variables, hidden states, observed state generation matrix and state transition matrix [24,25]. Fig 5 shows a detailed form.

In Fig 5, *Z* represents hidden variables, i.e. observable data. *X* represents an observational variable, which is data that cannot be observed directly. $Z_n$ is related to $Z_{n-1}$. $X_n$ is related to $Z_n$. The distribution of the model can be represented as shown in Eq (2) [26].

$$p(x_i, \cdots, x_N, z_1, \cdots, z_N) = p(z_1)[\prod_{n=2}^{N} p(z_n|z_{n-1})][\prod_{n=1}^{N} p(x_n|z_n)] \tag{2}$$

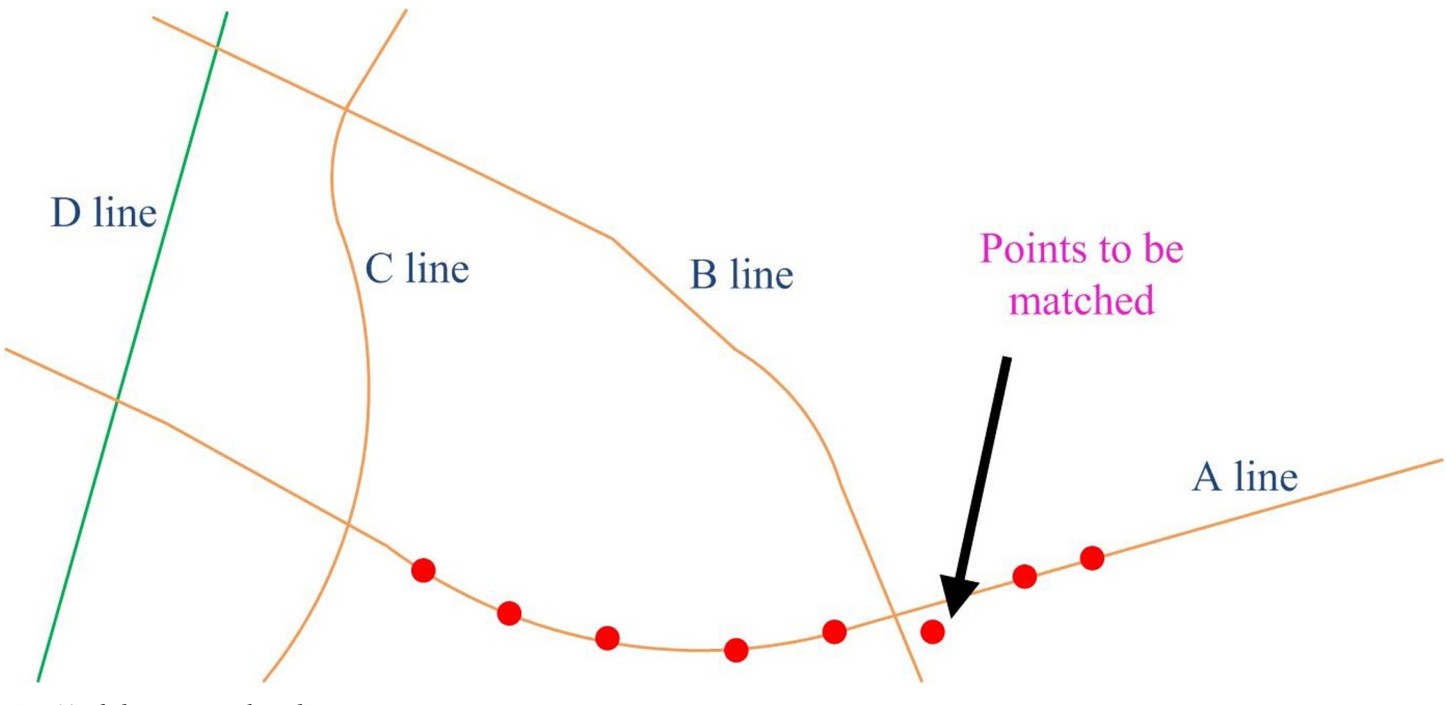

**Fig 4. Track data map matching diagram.**

In Eq (2), $p(x_i, \ldots, x_N, z_1, \ldots z_N)$ represents the sequence of observed variables, which represents the probability of $x_i, \ldots, x_N$ under the hidden variable $z_1, \ldots, z_N$. $p(z_1)$ indicates the probability of initializing hidden variables. $p(z_n|z_{n-1})$ represents the probability of transitioning from the hidden variable $z_{n-1}$ state to the state $z_n$. $p(x_n|z_n)$ represents the probability of the observed variable under the hidden variable $z_n$. By initializing the parameters of hidden variable probability, state transition probability, and observation variable generation probability, an HMM can be uniquely determined. Among them, the probability of initializing the hidden variable is assumed to be a random distribution $\theta$, as the variable does not have a direct precursor node. The normalization condition needs to be met as $\sum_{i=0}^{N} \theta_i = 1$, where there are $N$ different states. If a state $i$ is represented as $Z_1$, the probability distribution is represented as shown in Eq (3) [27–29].

$$p(Z_1|\theta) = \prod_{i=1}^{N} Z_i^{1i} \tag{3}$$

Assuming $Z_{n-1}$ is represented as the current state, each state can be represented by a vector for $K$, and each value in the vector only goes to (0, 1). So, the probability of generating the next state $Z_n$ is represented by the state transition matrix using the matrix $A$. The $j$ in the $i$ line represents the probability of the $K$ th element of state $Z_{n-1}$ taking $j$ as $Z_n$ under the premise of $i$. The distribution can be written as shown in Eq (4) [30].

$$p(z_n|z_{n-1}, A) = \prod_{j=1}^{K} \prod_{k=1}^{K} A_{j,k}^{Z_{n-1}j^{Z_{nk}}} \tag{4}$$

The observation state generation matrix represents the probability of transitioning from an implicit state to an observable value, $p(x_n|z_n,\emptyset)$, where one state is represented by the size of $K$.

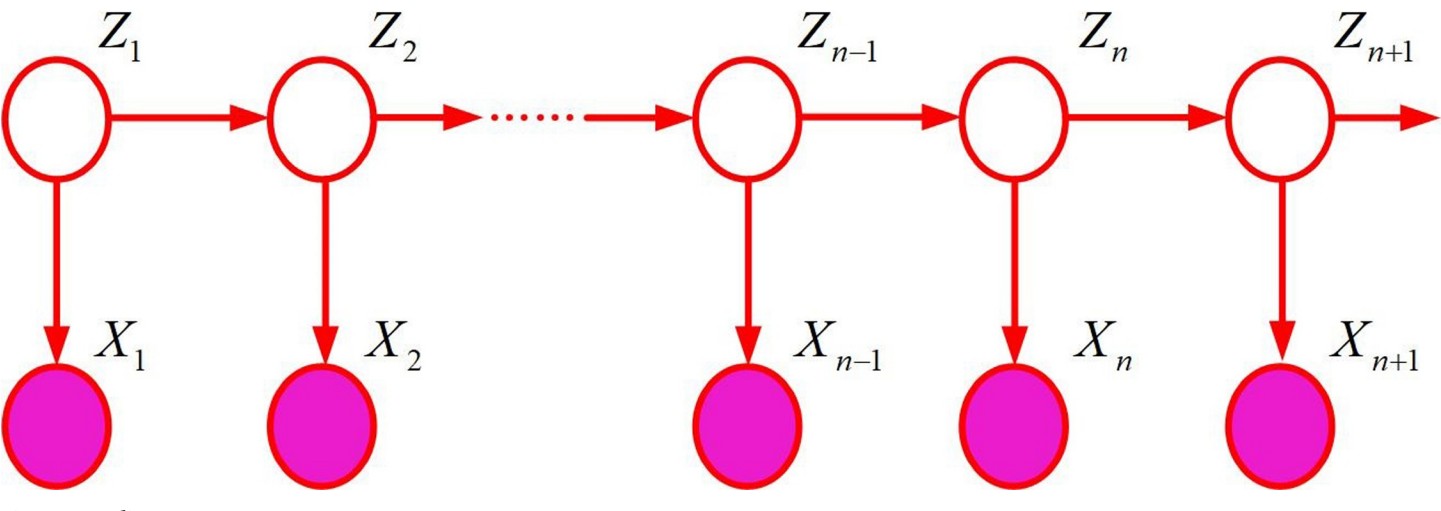

**Fig 5. HMM diagram.**

When $z_{nk} = 1$, it can represent the conditional probability at the $K$ value, as shown in Eq (5) [31].

$$p(x_n|z_n, \emptyset) = \prod_{k=1}^{K} p(x_n|\emptyset_k)^{z_{nk}} \tag{5}$$

## 3.2 Design of map matching algorithm based on the HMM

This study proposed a map matching algorithm based on HMM, aiming to solve this problem from a global perspective. Study set a parameter $K$ to select the closest line segments of $K$ to reduce computational complexity. It identified candidate points through these $K$ line segments and calculated the probability of generating observation states using a normal distribution. At the same time, a measurement method based on distance, velocity, and angle changes was proposed to construct a state transition matrix [32–34]. In the algorithm, the confirmation of candidate lines and candidate points under $K$ values was first studied. To achieve this goal, the study adopted a segmented cutting method. Specifically, whenever a break occurs in a line segment, this study will segment the entire network into multiple road segments and label each segment with a unique *id*. By using the pending point $p$, the point to line expression is as shown in Eq (6) [35].

$$d = \frac{|Ax_0 + By_0 + C|}{\sqrt{A^2 + B^2}} \tag{6}$$

In Eq (6), the line is represented as $Ax_0 + By_0 + C = 0$, where $A$, $B$, and $C$ are the parameters of the line. The first $K$ segments closest to the test point $p$ are selected as candidate segments [36,37], and the projection points from the processing point $p$ to each candidate segment are

calculated based on the selected candidate segments $e_1$, $e_2$ ... etc., as shown in Eqs (7–9) [38].

$$|BD| = BA \cdot \frac{BC}{|BC|} \tag{7}$$

$$BD = |BD| \cdot \frac{BC}{|BC|} = BC \cdot \frac{BA \cdot BC}{|BC|^2} \tag{8}$$

$$D = B + BD \tag{9}$$

In Eqs (7–9), $D$ represents the projection of the assumption on the $BC$ line segment, and $A$ represents the $p$ point. In the algorithm, the confirmation of the observation generation matrix is to determine the distribution relationship between the positioning accuracy and the real road. It is generally assumed that this difference follows a normal distribution and its variance is set to 1 [39,40]. The standard deviation can be determined using the method given in Eq (10) [41].

$$\mu = \frac{\sum_{k=1}^{K} [p_i - p_i^k]}{K} \tag{10}$$

In Eq (10), $i$ represents the point to be processed. $p_i^k$ represents the $k$ th candidate point of the $i$ th true point. $k$ represents the number of candidate points. It calculates the distance from each candidate point to that point, and takes the mean, as shown in Eq (11) [42].

$$N(c_i^j) = \frac{1}{\sqrt{2\pi}\sigma} e^{-\frac{(x_i^j - \mu)^2}{2\sigma^2}} \tag{11}$$

In Eq (11), $x_i^j$ represents the distance between the $i$ th point and the candidate point $j$. For the confirmation of the state transition matrix, this study proposes to start with the angle, velocity, and distance between the processing point $pt$ and its previous point $pt$—1. Firstly, the ratio of the distance between the processing point and its precursor candidate point to the distance between the positioning point can be determined using Eq (12) [43].

$$L(c_{i-1}^t \rightarrow c_i^s) = \frac{d_{i-1 \rightarrow i}}{w_{(i-1,t) \rightarrow (i,s)}} \tag{12}$$

In Eq (12), $i$ represents the current pending point. $i$—1 represents front wheel drive. $t$ and $s$ represent the determined candidate points, where $d$ distance and $w$ calculate the distance of the positioning point. It further combines speed changes to calculate the speed of candidate line segments. Specifically, the distance $d$ can be determined by the distance between the previous and subsequent candidate points, and the instantaneous velocity on the line segment can be calculated, as shown in Eq (13) [44].

$$\bar{v}_{(i-1,t) \rightarrow (i,s)} = \frac{d_u}{\Delta t_{i-1 \rightarrow t}} / v_{(i-1,t) \rightarrow (i,s)} \tag{13}$$

Finally, the similarity of angles can be combined to verify whether the change angles of the selected candidate points are similar. This can be achieved with Eq (14) [45].

$$r_{jk} = \frac{\sum_{i=1}^{n} x_{ij} x_{ik}}{(\sum_{i=1}^{n} x_{ij}^2 \sum_{i=1}^{n} x_{ik}^2)^{\frac{1}{2}}} \tag{14}$$

In summary, the similarity of distance, speed, and angle can be combined to calculate the

overall formula, as shown in Eq (15) [46].

$$F_t(c_{i-1}^t \rightarrow c_i^s) = L(c_{i-1}^t \rightarrow c_i^s) + \bar{v}_{(i-1,t)\rightarrow(i,s)} + 1/r_{jk} \tag{15}$$

Finally, it will normalize the results and calculate the state transition matrix. In this study, an algorithm is used to cut the line and the nearest $K$ different segments are selected to reduce computational overhead and determine candidate points for each segment. To achieve this goal, a state transition matrix and an observation generation matrix are constructed based on HMM. Through these matrices, this study can calculate results that accurately express specific line groups. The specific algorithm is shown in Table 2.

In Table 2, first is to input trajectory data, circuit diagram, and a specific value. Trajectory data is a series of positional information used to represent the movement path of a trajectory. Route maps are pre-prepared geographic information data such as road networks or bus routes. The selection of values is related to the accuracy and efficiency of matching. Next, for each trajectory data point, it will output the corresponding line segment group. Then, it will initialize the data and set an $K$ value to affect the accuracy and effectiveness of subsequent matching results. Subsequently, the overall route map is cut into fragments related to each trajectory data point. Finally, the probability of generating the trajectory data for all line segments is calculated. The above steps need to be repeated N times to process and match all trajectory data points.

## 3.3 Source of trajectory data information

According to the Gaia Trajectory Data Open Program, data was collected from partial areas of the Second Ring Road in Xi'an and Chengdu from October 2016 to November 2016. Table 3 provides specific information on these data. These data can be used to analyze information on traffic flow, travel patterns, and road utilization.

Table 3 shows the relevant data information of Gaia data. The city name represents the source city of the data. The schedule shows the time range of data collection. The data volume represents the size of the compressed data. The type represents the meaning represented by each data. The collection interval represents the time interval of data collection, usually 3 seconds. In a large amount of traffic trajectory data, there is often a lot of irrelevant information. Therefore, before conducting research, it is necessary to pre-process the data. The Chengdu Second Ring Road ride hailing Gaiya data is taken as an example, as shown in Table 4.

According to the attributes of ride hailing data in Table 4, each record contains the following information: vehicle ID, order information, time information, latitude and longitude information, start time, end time, the latitude and longitude of boarding and alighting. Among them, vehicle ID is used to identify each ride hailing vehicle. Order information is a randomly

**Table 2. K-HMM algorithm flow table.**

| Algorithm flow |
| --- |
| Input: track data, route diagram, K value |
| Output: the group of line fragments corresponding to each track data |
| Repeat the following steps N times: |
| (1) Initialize data and set K value |
| (2) Cut the overall circuit diagram. |
| (3) Determine the nearest K lines and candidate points by calculating formulas 6 and 9 |
| (4) Calculated the observation generation matrix and state transition matrix by formulas 11 and 15 |
| (5) Calculate the probability of generating the trajectory data for all medium line segments |

**Table 3. Data information table.**

| City name | Time | Data volume | Type | Acquisition interval |
|---|---|---|---|---|
| Xi 'an Second Ring Road | October 2016 | 5.5 | Tracking point | 2–4 |
| Chengdu Second Ring Road | October 2016 | 8.0 | Tracking point | 2–4 |
| Xi 'an Second Ring Road | November 2016 | 5.3 | Tracking point | 2–4 |
| Chengdu Second Ring Road | November 2016 | 10.1 | Tracking point | 2–4 |

generated string of symbols by the system that represents a unique customer order. The time information records the time at which each data is generated. The latitude and longitude information represents the specific location of the ride hailing service. The start and end time record the start and end times of the ride hailing service, respectively. The latitude and longitude of boarding and alighting record the positions of passengers boarding and alighting, respectively.

In Table 5, there are issues with missing, memory confusion, and redundancy in the original trajectory data. To address these issues, it is necessary to perform simple data processing. In the data pre-processing phase, the study first examined each record to confirm if there is any missing information. Then, the redundant data was processed to identify and delete all duplicate records with identical properties in the data set. Further, it should pay attention to the accuracy of the time logic. In addition, the positional logic was scrutinized to ensure that no physically impossible speeds or distances occur. Table 6 shows the processed data. Through these steps, more accurate trajectory data can be obtained.

## 4. Results

In this research experiment, the dataset used in this study is sourced from Section 3.3. For hardware environments, research used Intel Core i7-8700k processors and 16GB of memory. The operating system was Windows 10, and the software environment was Python 3.7 programming language for algorithm implementation and data processing. The development environment was Anaconda, the data analysis library was Pandas and NumPy, and the map matching algorithm library was HMM library.

**Table 4. Original network car properties sheet.**

| Attribute | Name | Example | Description |
|---|---|---|---|
| Vehicle ID | String | a739b90e4907fa30b0d6a3a3b39e67bb | Uniquely marked vehicle |
| Order ID | String | 982bf243c3202415d6252271b2693161 | Uniquely represents order information |
| Time | Time | 1478041356 | The time format is nix time cut, accurate to seconds |
| Longitude | Double | 104.10018 | The format is GCJ-02 |
| Dimensionality | Double | 30.70847 | The format is GCJ-02 |
| Start time | Time | 1478041359 | The time format is nix, accurate to second |
| End time | Time | 147804149 | The time format is nix, accurate to second |
| Boarding longitude | Double | 104.10018 | The format is GCJ-02 |
| Boarding dimension | Double | 30.70847 | The format is GCJ-02 |
| Alighting longitude | Double | 124.10057 | The format is GCJ-02 |
| Exit dimension | Double | 30.70908 | The format is GCJ-02 |

**Table 5. Part of the original data.**

| Vehicle ID | Order ID | Time | Longitude | Dimensionality | Start time |
|---|---|---|---|---|---|
| a739b90e4907fa | 982bf243da | 1478041353 | 104.10057 | 30.70908 | 1478041353 |
| a739b90e4907fa | 982bf244da | 1478041356 | 104.1005 | 30.70883 | 1478041353 |
| a739b90e4907fa | 982bf245da | 1478041359 | 104.10018 | 30.70847 | 1478041353 |
| a739b90e4907fa | 982bf246da | 1478041362 | 30.7082 | 104.2 | 1478041353 |
| a739b90e4907fa | 982bf247da | 1478041365 | 104.09992 | 30.70807 | 1478041353 |
| a739b90e4907fa | 982bf248da | 1478041368 | 104.09992 | 30.70807 | 1478041353 |
| a739b90e | 982bf248 | 1.5E+09 | 140.1 | 30.72 | 1.5E+09 |
| End time | Boarding longitude | Boarding dimension | Alighting longitude | Exit dimension | \ |
| 1478042363 | 104.10057 | 30.70908 | 104.10350 | 30.71446 | \ |
| 1478042363 | 105.10057 | 30.70908 | 105.10350 | 31.71446 | \ |
| 1478042363 | 106.10057 | 30.70908 | 106.10350 | 32.71446 | \ |
| 1478042363 | 107.10057 | 31.70908 | 107.10350 | 33.71446 | \ |
| 1478042363 | 108.10057 | 30.70908 | 108.10350 | 34.71446 | \ |
| 1478042363 | 109.10057 | 29.70908 | 109.10350 | 35.71446 | \ |
| 1.5E+09 | 110.20000 | 30.71 | 110.10349 | 36.72 | \ |

## 4.1 Trajectory data results and analysis of HMM-based map matching algorithm

In the experimental verification, the study used GPS trajectory data from the real world and employed screening methods and mapping formulas. In addition, the study also calculated the trajectory segments and generated corresponding state matrix results. The experimental data was sourced from the Gaia Trajectory Data Open Program, which selected local data from Chengdu Second Ring Road and preprocessed the data to remove redundant and missing data. $K$ selected candidate paths and candidate points are shown in Fig 6. From Fig 6(A), the fine-grained offset data appeared in a trajectory data, which required map matching work. Fig 6(B) shows the effect of calculating the $K$ line segment closest to the processing point according to the corresponding Eq 6, and the segment was numbered when $K$ is shown in the figure.

It was filtered through candidate paths, and then mapping formulas were utilized to find the mapping points between the processing points and each line. However, solely considering the distance between the processing point and the candidate point to determine which segment the processing point belongs to did not provide high positioning accuracy from a geometric perspective, as shown in Fig 7.

From Fig 7, in the map matching, only considering the geometric proximity of the nearest route and candidate points might lead to matching errors. From Fig 7(A) and 7(B), if candidate points were selected only based on geometric distance, then the calculated distance between the point to be processed and candidate point was closer than that of candidate point 3. Then candidate point 2 might be mistakenly replaced by the pre-processing point. From Fig

**Table 6. Retain a valid field property list.**

| Attribute | Name | Example | Description |
|---|---|---|---|
| Vehicle ID | String | a739b90e4907fa30b0d6a3a3b39e67bb | Uniquely marked vehicle |
| Order ID | String | 982bf243c3202415d6252271b2693161 | Uniquely represents order information |
| Time | Time | 1478041356 | The time format is nix time cut, accurate to seconds |
| Longitude | Double | 104.10018 | The format is GCJ-02 |
| Dimensionality | Double | 30.70847 | The format is GCJ-02 |

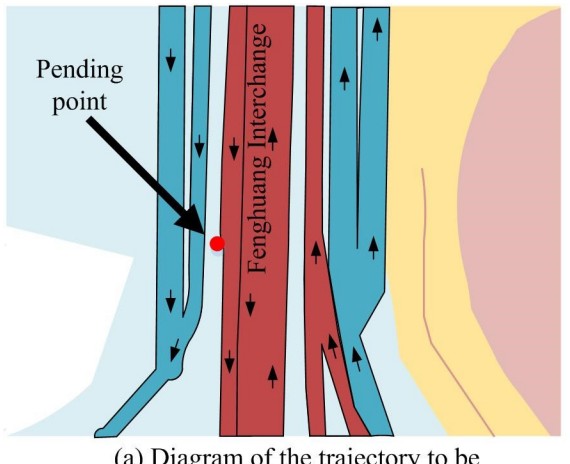

(a) Diagram of the trajectory to be processed

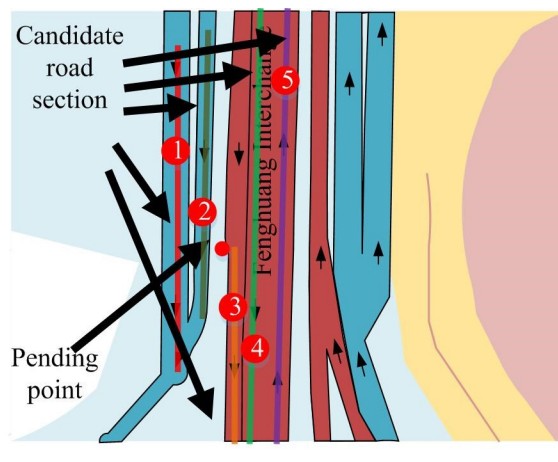

(b) Candidate path graph

**Fig 6. Candidate path display diagram.**

7(B), by adding other track points of the track data where the candidate point was located, the real candidate point of the candidate point should be candidate point 3. Therefore, in the map matching, only from the geometric point of view was not enough to ensure high precision results. This is similar to the research results of K. Researchers K. Zhang et al. [47] designed a spatio-temporal trajectory data compression algorithm that identifies the turning and speed change behavior of vehicles by analyzing their motion patterns, and extracts feature points from multiple perspectives.

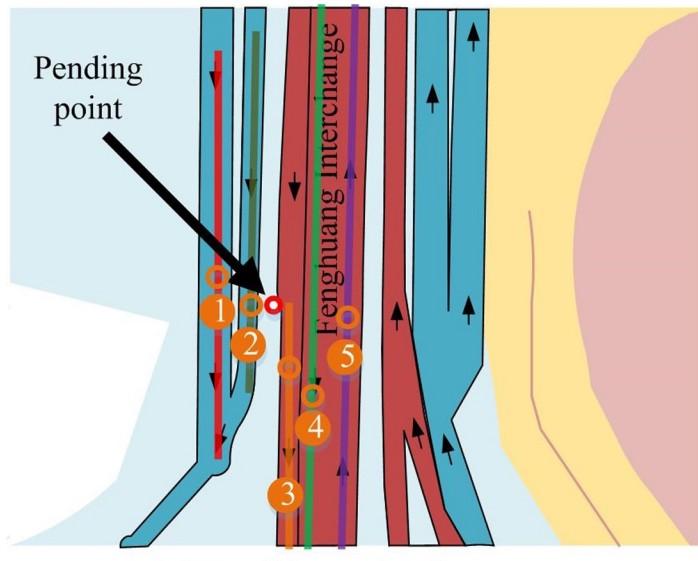

(a) Candidate point diagram

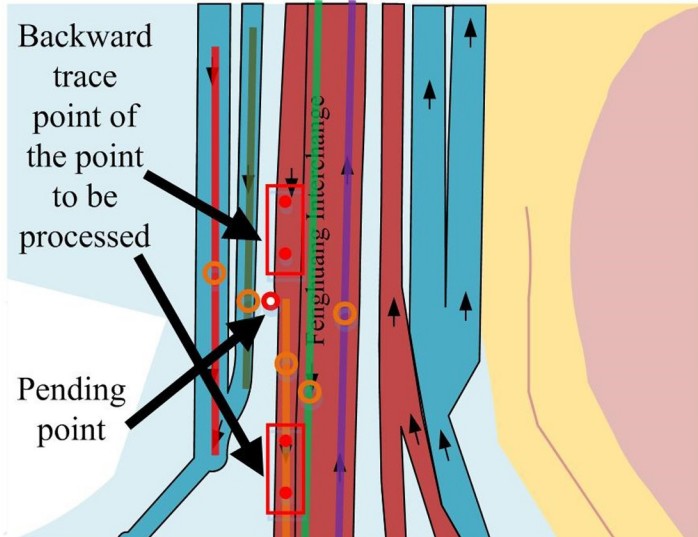

(b) Partial trajectory diagram

**Fig 7. Candidate points and partial trace points.**

## 4.2 Results and analysis of trajectory data after removing noise

According to the analysis results in the previous section, this study obtained a batch of data for map matching after removing noise in Section 2.1. The trajectory data after removing noise was local data from the Chengdu Second Ring Road in the preprocessed Gaia trajectory data open plan. It intended to test the superiority of K-HMM algorithm in map matching of trajectory data. In Fig 8, the trajectory data exhibited certain curves, variations, and intersections. This reflects the complexity and diversity of transportation networks in the real world.

Fig 8(A) shows the complex route map of Chongqing Second Ring Road, which shows the interweaving of traffic trajectories and presents a very complex situation, making it difficult to handle. Fig 8(B) shows a real ride hailing trajectory point. From the figure, due to factors such as the complexity of the transportation network, the trajectory points have shifted on various routes, which was not consistent with the real route. The algorithm proposed in this study used a $K$ value of 5 to calculate the trajectory segment, and the generated state matrix results are shown in Table 7.

In Tables 7 and 2 decimal places were retained, and each data was generated using a normal distribution. Its practical significance was that the closer the positioning point was to the candidate point, the closer its value was to 1. Among them, $p_i$ represents the trajectory point of the $i$ th positioning, and $i$ represents different points when taking different values. $c_j$ represents the $j$ th candidate point, while $j$ points to different candidate points with different values, with the highest value being $K$. Each $p_i$ corresponds to its own $c_j^i$ selection point, with the highest $j$ value taken as $K$. For the state transition matrix, it integrates information on velocity, angle changes, and distance. Therefore, the study used normalization methods to calculate the state transition matrix between candidate points $c_j^{i-1}$ corresponding to time $p_{i-1}$ of time $t_{i-1}$, and the state transition matrix of candidate point $c_j^i$ corresponding to time $p_i$ of time $t_i$. Fig 9 shows the relationship between all positioning points and their corresponding candidate points.

Fig 9 shows that the state transition matrix needs to calculate the state transition matrix between the candidate points in the previous moment and the candidate points in the next

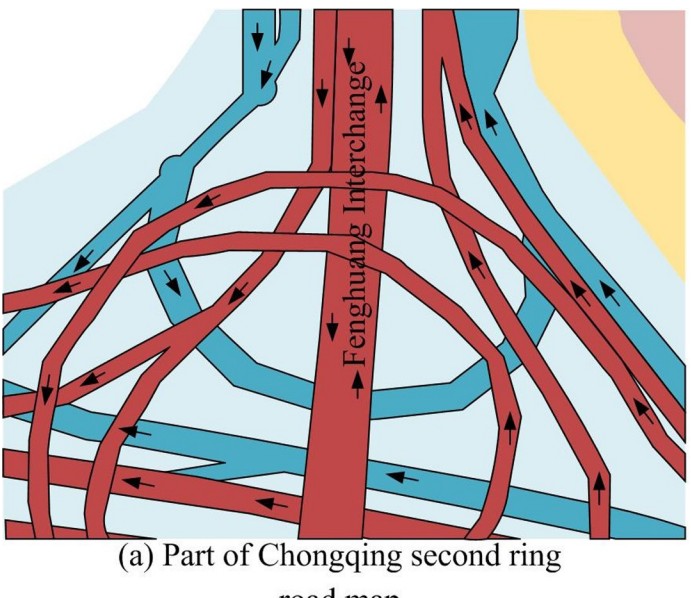

(a) Part of Chongqing second ring road map

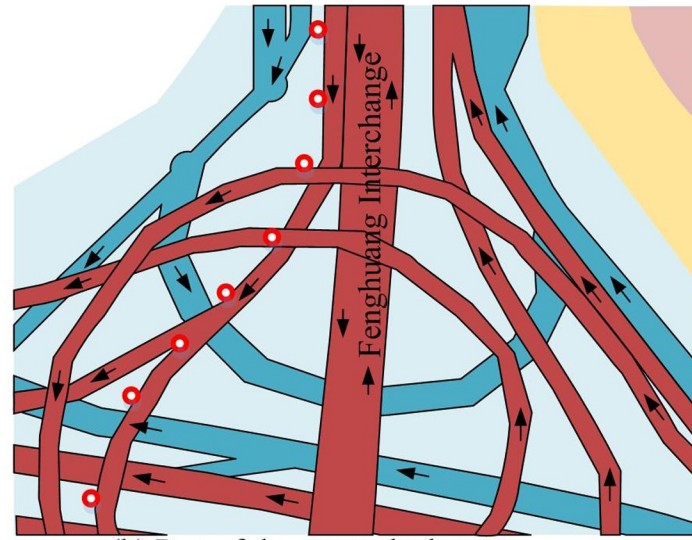

(b) Part of the network about car track point diagram

**Fig 8. Complex traffic line and track point diagram.**

**Table 7. State generation matrix table.**

| / | $c_1$ | $c_2$ | $c_3$ | $c_4$ | $c_5$ |
|---|---|---|---|---|---|
| $p_1$ | 0.84 | 0.56 | 0.42 | 0.44 | 0.37 |
| $p_2$ | 0.62 | 0.63 | 0.34 | 0.54 | 0.76 |
| $p_3$ | 0.59 | 0.54 | 0.76 | 0.13 | 0.62 |
| $p_4$ | 0.36 | 0.66 | 0.72 | 0.66 | 0.22 |
| $p_5$ | 0.12 | 0.4 | 0.25 | 0.69 | 0.55 |
| $p_6$ | 0.89 | 0.68 | 0.58 | 0.57 | 0.99 |
| $p_6$ | 0.7 | 0.79 | 0.36 | 0.25 | 0.28 |

moment. Table 8 shows the state transition table between the candidate points corresponding to $p_1$ to $p_2$.

As shown in Table 8, the state transition matrix table from $p_1$ to $p_2$ should retain two significant digits after the decimal point. It simply calculated the state transitions of all trajectory points to be processed from $p_2$ to $p_3$, $p_3$ to $p_4$, and combined the state generation matrix table with the state transition matrix table to calculate the combination of all candidate points. It selected the candidate combination with the largest data as the true matching path, as shown in Fig 10, which is a complex traffic route and trajectory point map.

Fig 10(A) shows the online ride hailing positioning data represented by red trajectory points, while yellow trajectory points represent results that only consider geometric relationships (neighboring points). From the figure, the nearest algorithm that only considered the distance between points and lines was prone to positioning errors. This is due to without considering the connectivity issue of the line and the temporal dimension characteristics of trajectory data, and

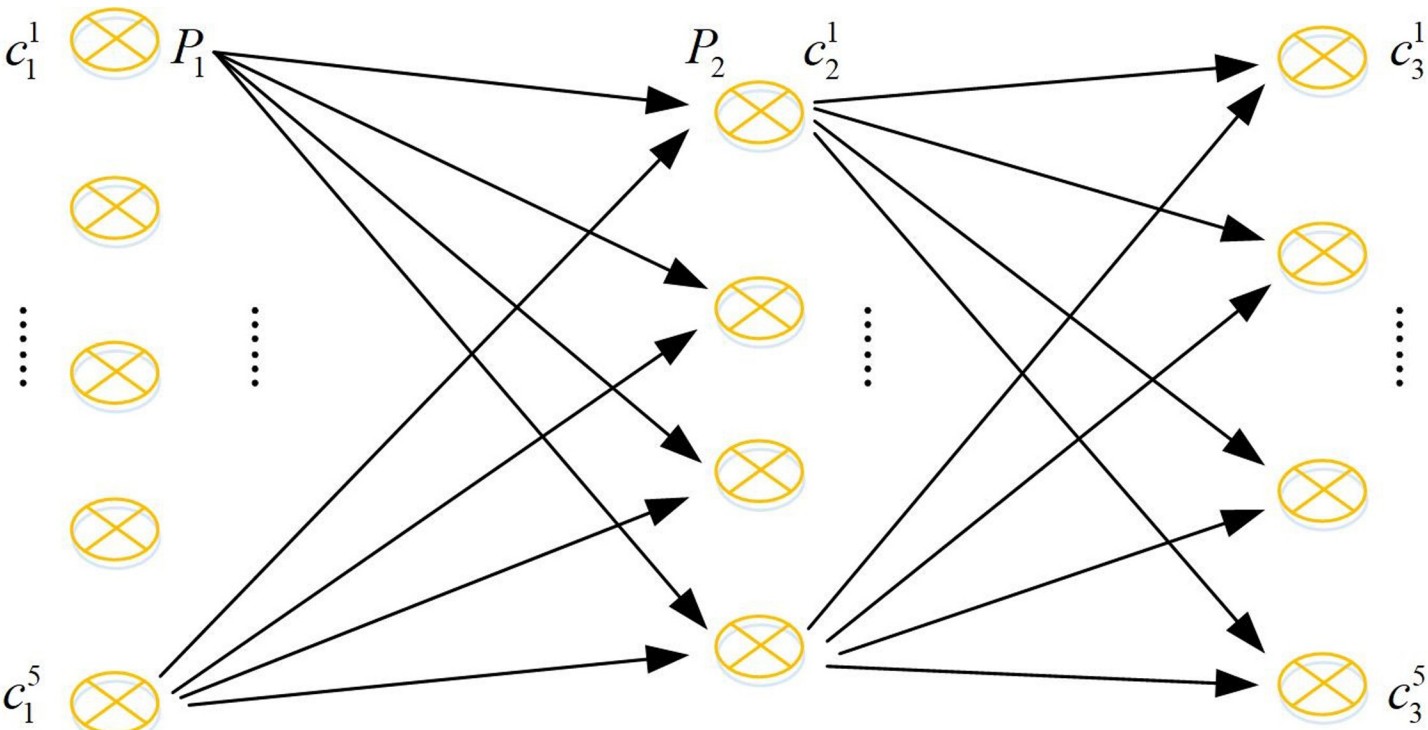

**Fig 9. The relationship diagram between the candidate points corresponding to the anchor points.**

**Table 8. State transition table.**

| / | $c_1^2$ | $c_2^2$ | $c_3^2$ | $c_4^2$ | $c_5^2$ |
|---|---|---|---|---|---|
| $c_1^1$ | 0.57 | 0.32 | 0.1 | 0.02 | 0.04 |
| $c_2^1$ | 0.24 | 0.36 | 0.21 | 0.12 | 0.12 |
| $c_3^1$ | 0.69 | 0.04 | 0.22 | 0.06 | 0.04 |
| $c_4^1$ | 0.13 | 0.03 | 0.72 | 0.09 | 0.08 |
| $c_5^1$ | 0.32 | 0.57 | 0.04 | 0.07 | 0.05 |

there was correlation between trajectory points. Fig 10(B) shows the processing results of the proposed algorithm. The red points represent the original positioning data points, while the yellow trajectory points represent the results processed by the algorithm. The proposed algorithm considered the average speed, distance, and angle of candidate paths as starting points, and studied the relationship between the two points before and after. Therefore, the accuracy has significantly improved. In this algorithm, the study involved selecting the nearest line, which requires a manually specified value. 10 to 20 independent experiments were performed for each $K$ value to ensure the reliability of the statistical results. This study was measured from the time and accuracy, and provided the appropriate value that guarantees accuracy and efficiency. The experimental results are shown in Table 9. When $K = 5$, the algorithm accuracy reached 95.3, and the calculation time was only 51ms. Compared with $K = 7$ (accuracy was 97.1, calculation was102ms), the calculation time was greatly reduced while maintaining high accuracy. This balance was based on consideration of the application scenarios of the algorithm. During the experiment, statistical analysis was also conducted. The study used chi square test to compare multiple sets of data pairwise and output a significance value $p$.

The time loss in Table 9 refers to the average time spent processing a trajectory point, compared to accuracy. From the results in the table, as the $K$ value changed, the average time to

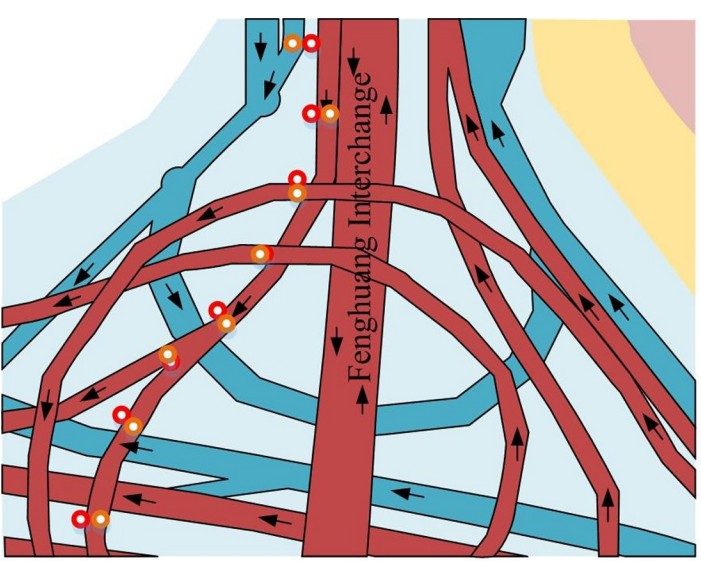

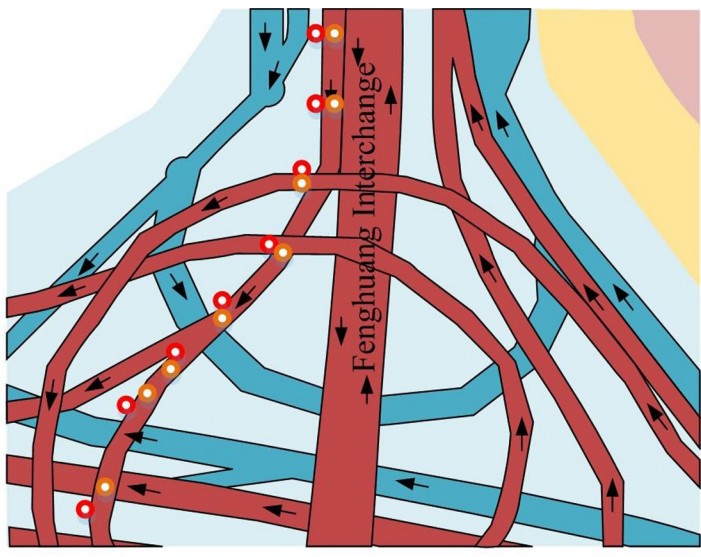

(a) The resulting graph after geometric relation processing

(b) Study the resulting graph after the proposed algorithm

**Fig 10. Complex traffic line and track point diagram.**

process a trajectory point continued to increase. However, when the K value reached a certain value, the increase in accuracy did not change much. Considering both time loss and accuracy, $K = 5$ was chosen as the optimal value for processing trajectory data. Through the above experimental results, the algorithm performed best when the $K$ value was set to 5. $p < 0.05$ in Table 9 indicates that the corresponding data was significant. The comparison of accuracy and time loss with other algorithms is shown in Fig 11.

Fig 11(A) shows that the algorithm proposed in this study performed well in various road conditions, with the best performance in parallel road sections and an accuracy of about 96% in mixed road sections. The accuracy of the algorithm designed by the research on parallel, crossing, overpass, and mixed sections was 98.3%, 97.5%, 94.8%, and 96%, respectively, with an average value of 96.65%. The accuracy of the traditional HMM algorithm was 95.9%, 95.7%, 95.4%, and 94.6%, with an average of 95.4%. The accuracy of the direct projection algorithm was 95.2%, 93.1%, 90.4%, and 84.4%, respectively, with an average value of 90.775%. The accuracy of curve fitting was 96.3%, 94.9%, 94.2%, and 90.8%, respectively, with an average value of 94.05%. Fig 11(B) shows the average time loss of various algorithms in processing positioning points. From the figure, the time loss of the proposed algorithm was reduced, with a time loss of nearly 60ms in the overpass section. The time loss of the proposed algorithm on parallel, crossing, overpass, and mixed sections was 53.1ms, 55.9ms, 60ms, and 58.8ms, respectively, with an average value of 56.95ms. The time loss based on traditional HMM algorithm was 53.8ms, 57.5ms, 68.9ms, and 62.8ms, with an average value of 60.75ms. The time loss of the direct projection algorithm was 23.5ms, 32.1ms, 42.6ms, and 39.7ms, respectively, with an average value of 34.475ms. The time loss for curve fitting was 27.6ms, 38.3ms, 48.2ms, and 46.3ms, respectively, with an average value of 40.1ms. Overall, although the algorithm increased time loss compared to algorithms based on geometric principles, its accuracy performance was very stable. In summary, a detailed explanation of the map matching algorithm in this study was provided. The final results of using the K-HMM algorithm on Chengdu data are shown in Fig 12.

## 5. Discussion

A map matching algorithm based on HMM was studied and designed to address the issue of deviation between positioning data and real roads in traffic trajectory data analysis. This algorithm not only introduced HMM, but also involved considering geometric shapes and combining time series information of orbital points. The results showed that the map matching algorithm based on HMM had good performance, with good accuracy and time loss. The accuracy was about 96% in mixed road sections, and it reduced nearly 60 milliseconds in overpass sections. This result was consistent with existing research on HMM. F. Liu et al. [48] designed a Gaussian HMM for the regionalized decision-making problem of human-machine

**Table 9. A table of time loss and accuracy under different K values.**

| *I* | Time loss (ms) | Accuracy rate (%) | *p* |
|---|---|---|---|
| $K = 1$ | 9 | 78.7 | 0.031 |
| $K = 2$ | 13 | 79.6 | 0.027 |
| $K = 3$ | 33 | 85.7 | 0.023 |
| $K = 4$ | 46 | 89.0 | 0.034 |
| $K = 5$ | 51 | 95.3 | 0.022 |
| $K = 6$ | 90 | 97.0 | 0.019 |
| $K = 7$ | 102 | 97.1 | 0.017 |

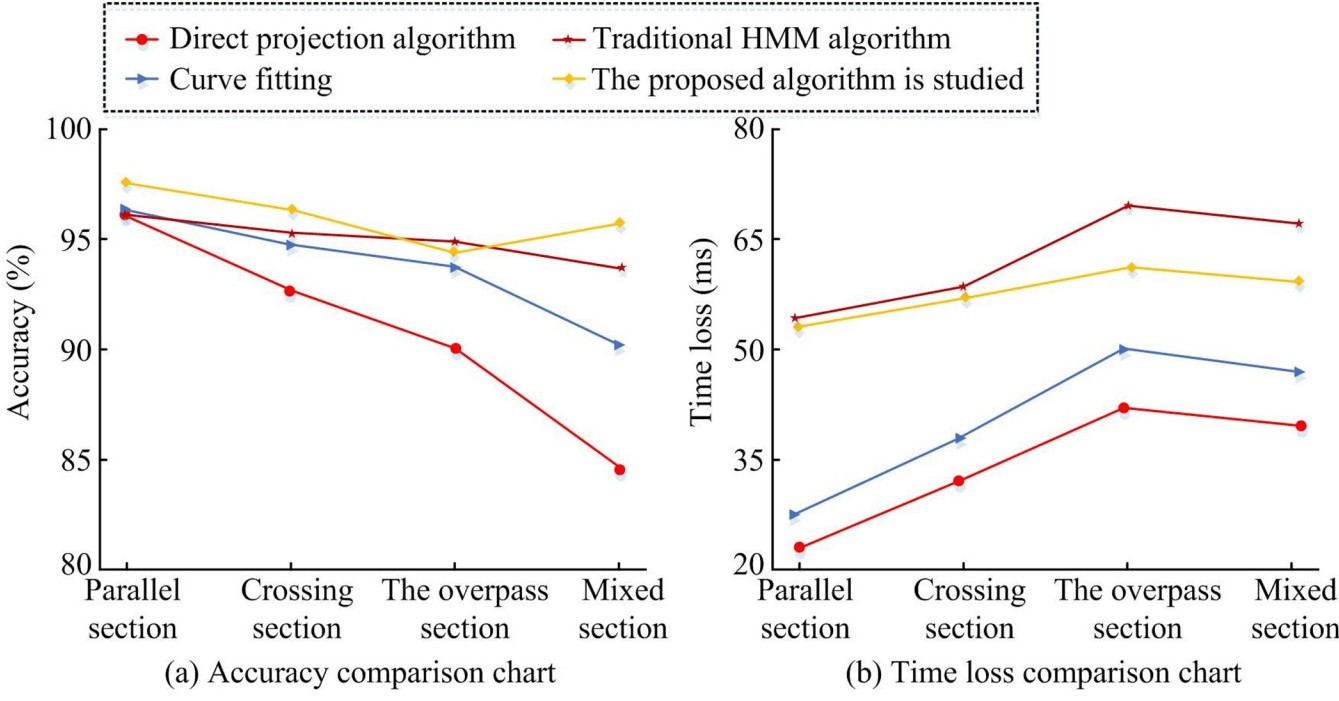

**Fig 11. Algorithm comparison graph.**

shared control. The results showed that the model performed well in both accuracy and time loss, and could adaptively adjust the driver's control switching.

## 6. Conclusion

This study introduced the source and composition of data, and discussed pre-processing methods for trajectory data, as well as map matching algorithms for processing biased data. Due to the inconsistency between the data obtained by the positioning tool and the actual path, path matching work needed to be carried out. TMM techniques only considers geometric information, resulting in low accuracy. Therefore, a map matching algorithm based on HMM was proposed. From a global perspective, $K$ candidate paths and points were selected, and an observation variable generation matrix was constructed using the idea that the accuracy error generally follows a normal distribution. The state transition matrix was determined by fusing the speed of the candidate road segment, the distance from the positioning data, and the angle information to complete the map matching work of the entire algorithm. The experiment selected a $K$ value of 5 as the optimal value to process trajectory data, with a time loss of 51 ms and an accuracy of 95.3. The proposed algorithm performed well, with the best performance in parallel road sections and an accuracy of about 96% in mixed road sections. The time loss was also reduced, with nearly 60 ms in the overpass section. Although the algorithm proposed in the study increased time loss compared to algorithms based on geometric principles, its accuracy performance was very stable. However, there are shortcomings in this study. Due to data collection conditions and cost limitations, the scale of traffic trajectory data used in the study is relatively small. When processing deviation data, its accuracy needs to be further improved, and more suitable means are needed to extract the information vector of trajectory data points. This is also an area where further research can continue to improve.

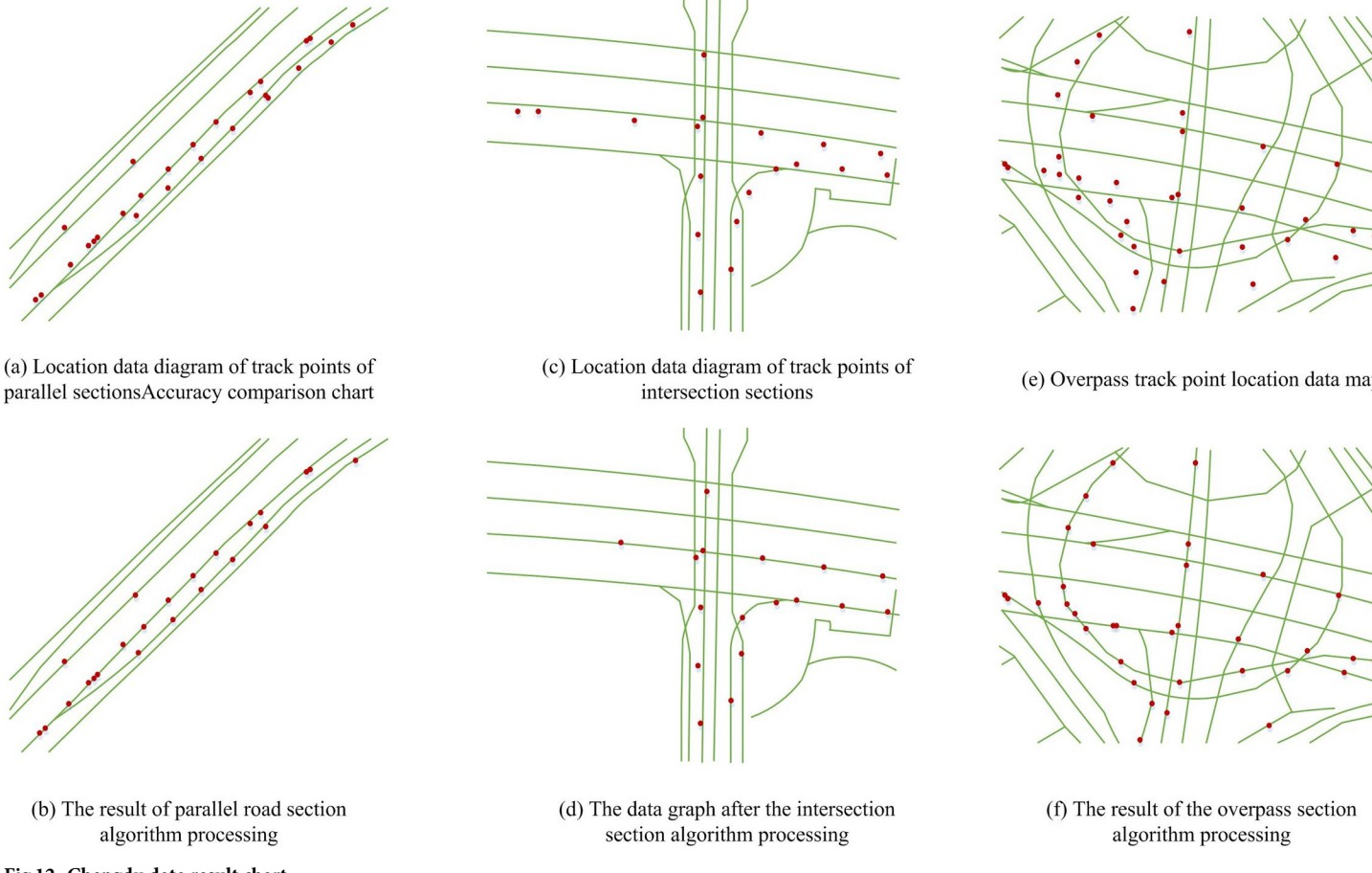

(a) Location data diagram of track points of parallel sectionsAccuracy comparison chart

(c) Location data diagram of track points of intersection sections

(e) Overpass track point location data map

(b) The result of parallel road section algorithm processing

(d) The data graph after the intersection section algorithm processing

(f) The result of the overpass section algorithm processing

**Fig 12. Chengdu data result chart.**

## Supporting information

**S1 Dataset.**
(DOC)

## Author Contributions

**Data curation:** Mingkang Sun.

**Formal analysis:** Mingkang Sun.

**Investigation:** Mingkang Sun.

**Methodology:** Xiang Li.

**Software:** Xiang Li.

**Writing – original draft:** Xiang Li.

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
