## [Decision Letter · Decision Letter 0]

24 Jan 2024

PONE-D-23-41514Traffic Trajectory Data Analysis and Key Technologies of Map Matching Algorithm Based on HMM ModelPLOS ONE

Dear Dr. Sun,

Thank you for submitting your manuscript to PLOS ONE. After careful consideration, we feel that it has merit but does not fully meet PLOS ONE’s publication criteria as it currently stands. Therefore, we invite you to submit a revised version of the manuscript that addresses the points raised during the review process.

We look forward to receiving your revised manuscript.

Kind regards,

Yan Wang

Academic Editor

PLOS ONE

Reviewers' comments:

Reviewer's Responses to Questions

**Comments to the Author**

1. Is the manuscript technically sound, and do the data support the conclusions?

Reviewer #1: Partly

Reviewer #2: Yes

2. Has the statistical analysis been performed appropriately and rigorously? 

Reviewer #1: No

Reviewer #2: Yes

3. Have the authors made all data underlying the findings in their manuscript fully available?

Reviewer #1: No

Reviewer #2: No

4. Is the manuscript presented in an intelligible fashion and written in standard English?

Reviewer #1: No

Reviewer #2: Yes

5. Review Comments to the Author

Reviewer #1: The authors have submitted the manuscript " Traffic Trajectory Data Analysis and Key Technologies of Map Matching Algorithm Based on HMM Model". Although the manuscript can benefit the readers of the journal, but there are many issues with the manuscript in its current form. Some of them have been mentioned below:

1. The manuscript requires english editing for the audience to comprehend the text. There are many issue with writing of this manuscript. I would suggest the authors to ask native english speaker/editor to correct and proofread the manuscript.

2. I think the authors must revisit and revise the title, it should be focussing on Map matching algorithm using HMM model

3. The formation of sentences is complex, so must be simplified. There are many instances of it, Point 1 will help authors. The first 4-5 lines in abstract needs to be rewritten.

4. Section Introduction needs to re-written to add the required references. The statements has been made without proper citations. For example ... Traditional map matching algorithms....

Words are connected and needs to separated.

5. Section Related work (should be section 2) is one of the poorest in this manuscript. No sentence formation is incorrect (should be past tense) and does not seem to be connected with the flow, reference numbers are missing when citing authors names. Second last paragraph seems like random reference text, should be corrected or removed with similar context. The word, ...All in all .... is non-technical.

6. Third section should be Methodology that include HMM-based algorithm... Need to rewrite sentence here too. Figure 1 is actually a table. Remove duplicated statements in this section, subtitles are needs revision (should be short and crisp); Formulas are used without citations and need flow in the discussion.

7 Section 3 in the manuscript should be section 4 results. Again the issue with the wording confusing statements are mentioned.

Reviewer #2: This study proposes a map matching algorithm based on hidden Markov models. The proposed algorithm starts from the global route, selects K nearest candidate paths, and identifies candidate points through the candidate paths. It generates a state transition matrix using changes in speed, angle, and other information to match trajectory points to the actual route. In this paper, a key technology of map matching algorithm is proposed. The topic is interesting. However, there are some problems, which must be solved before it is considered for publication.

1. The abstract section discusses that the algorithm used in the paper is more accurate than traditional algorithms, but the running time will increase. This section only provides a simple description but lacks data comparison so it is not convincing enough. It is suggested that a detailed analysis of the advantages and disadvantages of the two methods in the "Introduction" section be added, which will demonstrate whether the map matching algorithm based on hidden Markov models can solve problems that traditional algorithms cannot solve in practical applications.

2. In the "HMM based map matching algorithm" section, the article mentions the need to preprocess data containing a large amount of attribute information, especially for data with information loss, redundant data, and incorrect information logic. However, no specific processing standards were mentioned. Therefore, you need to supple standards for data processing in this section.

3. According to the K-HMM algorithm flowchart in Table 4, is the value of T here the same as the track data T in Chapter 2.2? How to determine the value of T?

4. How many experiments are conducted for each K value in Table 7 to determine the time consumption and accuracy? Is it universal?

5. When explaining the geometric graph matching algorithm, the paper mentions that the trajectory points based on geometric factors are located on line B, while the true matching points are on line A. This part only explains the results without describing the process.

6. In Chapter 3.1, the Figure 8(b) “Partial trajectory map” was not explicitly used in the text. Perhaps it is to express the distribution of the remaining points when 3 points are used as trajectory candidate points, but there is no clear reference in the article.

6. PLOS authors have the option to publish the peer review history of their article (what does this mean?). If published, this will include your full peer review and any attached files.

Reviewer #1: **Yes: **Sukhjit Singh Sehra

Reviewer #2: No

---

## [Author Response · Author response to Decision Letter 0]

3 Mar 2024

The manuscript has been modified according to comments.

---

## [Decision Letter · Decision Letter 1]

17 Mar 2024

PONE-D-23-41514R1Traffic Trajectory Data Analysis Technology Based on HMM Model Map Matching AlgorithmPLOS ONE

Dear Dr. Sun,

Thank you for submitting your manuscript to PLOS ONE. After careful consideration, we feel that it has merit but does not fully meet PLOS ONE’s publication criteria as it currently stands. Therefore, we invite you to submit a revised version of the manuscript that addresses the points raised during the review process.

We look forward to receiving your revised manuscript.

Kind regards,

Yan Wang

Academic Editor

PLOS ONE

Journal Requirements:

Additional Editor Comments:

Authors should polish language and update references

Reviewers' comments:

Reviewer's Responses to Questions

**Comments to the Author**

1. If the authors have adequately addressed your comments raised in a previous round of review and you feel that this manuscript is now acceptable for publication, you may indicate that here to bypass the “Comments to the Author” section, enter your conflict of interest statement in the “Confidential to Editor” section, and submit your "Accept" recommendation.

Reviewer #1: (No Response)

Reviewer #2: All comments have been addressed

2. Is the manuscript technically sound, and do the data support the conclusions?

Reviewer #1: Yes

Reviewer #2: Yes

3. Has the statistical analysis been performed appropriately and rigorously? 

Reviewer #1: No

Reviewer #2: Yes

4. Have the authors made all data underlying the findings in their manuscript fully available?

Reviewer #1: No

Reviewer #2: Yes

5. Is the manuscript presented in an intelligible fashion and written in standard English?

Reviewer #1: Yes

Reviewer #2: Yes

6. Review Comments to the Author

Reviewer #1: The authors have submitted the manuscript " Traffic Trajectory Data Analysis Technology Based on HMM Model Map Matching Algorithm". The manuscript can benefit the readers of the journal. The authors response to the corrections made in not comprehensive, how they have improved the manuscript and what sections/lines have been updated.

I appreciate the authors for addressing inputs shared, but the English is still needs to be improved for making this manuscript publishable. Also, authors need to see how the related work in other articles is mentioned and how references are cited. It looks like discontinuity in the paragraphs of the manuscript. I would suggest to make the manuscript little short to say the same thing in fewer sentences.

Reviewer #2: (No Response)

7. PLOS authors have the option to publish the peer review history of their article (what does this mean?). If published, this will include your full peer review and any attached files.

Reviewer #1: No

Reviewer #2: No

---

## [Author Response · Author response to Decision Letter 1]

9 Apr 2024

The manuscript has been modified according to comments.

---

## [Editor Report · Decision Letter 2]

10 Apr 2024

Traffic Trajectory Data Analysis Technology Based on HMM Model Map Matching Algorithm

PONE-D-23-41514R2

Dear Dr. Sun,

We’re pleased to inform you that your manuscript has been judged scientifically suitable for publication and will be formally accepted for publication once it meets all outstanding technical requirements.

Kind regards,

Yan Wang

Academic Editor

PLOS ONE

Additional Editor Comments (optional):

the current version is suitable for publication.
---

## [Editor Report · Acceptance letter]

26 Apr 2024

PONE-D-23-41514R2 

PLOS ONE

Dear Dr. Sun, 

I'm pleased to inform you that your manuscript has been deemed suitable for publication in PLOS ONE. Congratulations! Your manuscript is now being handed over to our production team.

Kind regards, 

on behalf of

Dr. Yan Wang 

Academic Editor

PLOS ONE